# Comprehensive Genomic Profiling in the Management of Ovarian Cancer—National Results from Croatia

**DOI:** 10.3390/jpm12071176

**Published:** 2022-07-19

**Authors:** Dora Čerina, Višnja Matković, Kristina Katić, Ingrid Belac Lovasić, Robert Šeparović, Ivana Canjko, Žarko Bajić, Eduard Vrdoljak

**Affiliations:** 1Department of Oncology, University Hospital Center Split, School of Medicine, University of Split, 21 000 Split, Croatia; dora09cerina@gmail.com; 2Department of Gynecologic Oncology, University Hospital Center Zagreb, 10 000 Zagreb, Croatia; visnja.matko@gmail.com (V.M.); katic0kristina@yahoo.com (K.K.); 3Department of Radiotherapy and Oncology, University Hospital Center Rijeka, 51 000 Rijeka, Croatia; iblovasic@gmail.com; 4Department of Medical Oncology, Division of Medical Oncology, University Hospital for Tumors, Sestre Milosrdnice University Hospital Center, 10 000 Zagreb, Croatia; robertseparov@gmail.com; 5Department of Radiotherapy and Oncology, University Hospital Center Osijek, 31 000 Osijek, Croatia; dr.ivanamarkovic@gmail.com; 6Research Unit “Dr. Mirko Grmek”, University Psychiatric Hospital “Sveti Ivan”, Jankomir 11, 10 000 Zagreb, Croatia; zarko@biometrika.hr

**Keywords:** advanced ovarian cancer, comprehensive genomic profiling, targeted therapy, precision medicine

## Abstract

Today, in the era of precision medicine, the determination of genomic instability or other potentially targetable mutations, along with BRCA 1 and BRCA 2, is a crucial component of the diagnosis and treatment management of advanced ovarian cancer. Advanced technologies such as next-generation sequencing (NGS) have enabled comprehensive genomic profiling (CGP) analysis to become more feasible for routine use in daily clinical work. Here, we present the results for the first two years of an analysis of patients with advanced ovarian cancer on a national level. The aim was to establish the position of CGP in the daily clinical practice of treating ovarian cancer. We performed a multicenter, retrospective, cross-sectional analysis on the total population of Croatian patients who were newly diagnosed with locally advanced or metastatic ovarian cancer or whose initial disease had progressed from 1 January 2020 to 1 December 2021, and whose tumors underwent CGP analysis. All 86 patients (100%) analyzed with CGP had at least one genomic alteration (GA). The median LOH was 14.6 (IQR 6.8–21.7), with 35 patients (41%) having an LOH ≥ 16. We found BRCA-positive status in 22 patients (26%). Conventional testing, which detects only BRCA mutations, would have opted for therapy with PARP inhibitors in 22 (26%) of our patients. However, CGP revealed the need for PARP inhibitors in 35 patients (41%). The results identified a significantly higher number of women who would achieve a possible benefit from targeted therapy. Hence, we believe that CGP should be a backbone diagnostic tool in the management of ovarian cancer.

## 1. Introduction

Ovarian cancer is the eighth most common cancer diagnosed among women worldwide. While it usually occurs in women of older age, a significant number of patients are diagnosed at a younger age (≤55 years), especially women with a family history of ovarian cancer. Furthermore, when defining public health importance, more than 70% of women are diagnosed with locally advanced or metastatic disease with an expected 5-year survival rate of less than 30% [1]. Due to its obscure clinical presentation, diagnosis at advanced stages, and high mortality rate, ovarian cancer is the most lethal cancer of the female reproductive system and thus represents one of the hot topics in oncology with the need for significant advances in treatment. The last significant breakthrough in terms of chemotherapy administration occurred with the introduction of paclitaxel and carboplatin regimens at the end of 1990 [2]. Unfortunately, the introduction of immunotherapy directed against VEGF in combination with chemotherapy and as a maintenance treatment did not affect overall survival, despite a significant effect on progression-free survival [3,4,5]. Finally, targeted therapy with PARP inhibitors in patients with germline or somatic BRCA mutations has revolutionized therapy, statistically and clinically improving outcomes and increasing patient and societal expectations [6,7,8]. Since the introduction of the latter treatment, the determination of germline and somatic BRCA 1 and BRCA 2 status is mandatory in the diagnostic workup [9]. Additionally, in 2020, PARP inhibitors were approved for the treatment of ovarian cancer in patients with an established homologous recombination deficiency (HRD) status through BRCA or mutation of other genes involved in the HRD process [10]. HRD and consequent loss of heterozygosity (LOH), which represents the percentage of the tumor genome with a focal loss of one allele, lead to genomic instability and occur due to genetic or epigenetic inactivation of one or more HR pathway proteins, including BRCA 1, BRCA 2, RAD51C, ATM, PALB2, and BRIP1 [11,12,13]. A clinically significant LOH score with approved PARP inhibitor therapy was determined at a cut-off of ≥16 [8]. On the basis of these findings, a shift in the paradigm of the approach for diagnosing ovarian cancer is occurring, with molecular classification surpassing simple histological classification into type I and type II ovarian cancer, and targeted therapy is becoming the mainstay treatment for locally advanced or metastatic disease [14]. Thus, a determination of genomic instability or other potentially targetable mutations, along with BRCA 1 and BRCA 2, is a crucial component of the diagnosis and treatment management in these patients. Advanced technologies such as next-generation sequencing (NGS) are becoming more feasible and are used in daily clinical work. NGS provides insights into all exons of cancer-related genes and identifies four main classes of genomic alterations: base substitutions, insertions or deletions, gene rearrangements, and copy number variations. In addition to the main genomic alterations, CGP assays also determine their patterns and provide information regarding genomic instability or so-called “genomic scarring” by detecting tumor mutational burden (TMB), microsatellite instability (MSI), and loss of heterozygosity (LOH) through complex computational analysis. Today, in the era of precision oncology and following the expansion of targeted therapy and immunotherapy, several CGP assays have been approved by the U.S. Food and Drug Administration (FDA) for diagnostic, prognostic, and therapeutic purposes, one of which is FoundationOneCDx (Foundation Medicine Inc., Cambridge, MA, USA) [15,16,17]. CGP is becoming more available and widely used, but the question of its accurate applicability, utility, and cost benefit remains.

Ovarian cancer represents one of the major health burdens in Croatia due to its high mortality-to-incidence ratio (0.67), which puts Croatia among the countries with the highest mortality and incidence in Europe [18,19]. A potential reason for the high mortality-to-incidence ratio lies in the late diagnosis and lack of proper treatments. For instance, in 2018, Croatia was one of the countries with the lowest tier for PARP inhibitor uptake [19]. Ovarian cancer patients are treated with standard chemotherapy following surgery (or before when neoadjuvant therapy is indicated) or, in the case of initially metastatic disease, with platinum-based chemotherapy and paclitaxel every three weeks or dose-dense, +/− bevacizumab, or, recently, with PARP inhibitors, depending on the residual disease and BRCA status as well as the response to platinum therapy. In the treatment of recurrent disease, patients are also treated with standard chemotherapy based on platinum sensitivity, along with bevacizumab or with PARP inhibitors in cases of BRCA mutation. PARP inhibitors are given after response to platinum-based chemotherapy as a maintenance treatment. However, they are not indicated in cases of HRD or LOH as they are in some other European countries.

At the end of 2019, a CGP analysis of the tumor specimens provided by Foundation Medicine Inc. (FMI) began in Croatia for patients diagnosed with metastatic disease as a part of the project for the development and implementation of precision oncology on a national level in Croatia [20]. The first results and experiences are in the process of being analyzed, with the recently proven utility of CGP analysis in patients with metastatic uterine cancer [21].

As no established screening method is available, and the majority of women are consequently diagnosed in advanced stages with low survival rates, the diagnostic workup should receive special attention, particularly because all patients should have equal opportunities to be treated the same with already approved targeted therapies. Due to the CGP analysis provided by FMI, we present the results of the first two years of testing patients with locally advanced or metastatic ovarian cancer on a national level. The aim of this study was to present the number of patients with targetable BRCA 1 and BRCA 2 mutations compared with the total number of patients whose CGP results revealed a need for targeted therapy with PARP inhibitors, as well as other potential targeted treatments, and to establish the position of the CGP of ovarian cancer in daily clinical practice.

## 2. Materials and Methods

### 2.1. Project Design

We performed a multicenter, retrospective, cross-sectional analysis of the total population of Croatian patients who were newly diagnosed with locally advanced or metastatic ovarian cancer or whose initial disease had progressed from 1 January 2020 to 1 December 2021, and whose tumors underwent CGP analysis. The analysis was performed through FoundationOneCDx for all patients and was conducted in a laboratory certified by the Clinical Laboratory Improvement Amendments and the College of American Pathologists (Foundation Medicine Inc., Cambridge, MA, USA) [15,16,17]. The obtained tumor specimen was sampled from surgery or biopsy of the primary disease or metastases. Formalin-fixed, paraffin-embedded tissue was sent as a block and one slide stained with hematoxylin and eosin or 10 unstained slides with one slide stained with hematoxylin and eosin. The minimum surface area was 25 mm^2^, and the minimum tumor content was 20%, while the optimum was 30% of tumor nuclei, defined as the number of tumor cells divided by the total number of all cells with nuclei. Once the DNA was extracted, 50–1000 ng underwent whole-genome shotgun library construction and hybridization-based capture in order to detect alterations of 324 genes in total: 309 exons related to tumors, one promoter region, one non-coding RNA, and certain regions of introns in 34 frequently rearranged genes in tumors. Illumina^®^ HiSeq 4000 was used to sequence hybrid capture-selected libraries to a high uniform depth. The typical median depth of coverage was >500×, with >99% of exons at coverage of >100×. The sequenced regions were analyzed for four different types of alterations: base substitution, deletion or insertion, copy number variation, and gene redistribution in a group of genes associated with tumor development. The microsatellite status was based on genome-wide analysis of 95 microsatellite loci; TMB was determined by counting all synonymous and non-synonymous variants present at a 5% allele frequency or higher, with the total number presented as mutations per megabase (Muts/Mb) unit; and homologous recombination repair (HRR) mechanism was assessed for mutations in the 14 HRR genes, namely ATM, BARD1, BRCA1, BRCA2, BRIP1, CDK12, CHEK1, CHEK2, FANCL, PALB2, RAD51B, RAD51C, RAD51D, and RAD54L [17]. Depending on the results, patients were potentially administered CGP-guided therapy in accordance with the approved (on-label) standard treatment of care available in Croatia.

This real-world analysis was conducted in five Croatian institutions: University Hospital Centre Split, University Hospital Center Zagreb, the Clinic for Tumors Sestre Milosrdnice, and the University Hospital Centers in Rijeka and Osijek. The project was approved by the ethics committees of all participating institutions. Informed consent was obtained from all patients for the CGP analysis and data collection and publication. The data file was anonymized before the analysis, and the project was performed in accordance with the World Medical Association Declaration of Helsinki of 1975 as revised in 2013 [22].

In accordance with the journal’s guidelines, we will provide our data for the reproducibility of this study in other centers if such is requested.

### 2.2. Participants

We planned to include the entire population of patients who fulfilled the CGP criteria defined by the Croatian Oncology Society: sufficient tissue for the CGP, good general health (ECOG performance status ≤ 2), and at least 12 months of life expectancy [20]. Hence, we did not perform a power analysis before starting the project. Patients were administered the first-line standard of care treatment for locally advanced or metastatic ovarian cancer, depending on their general condition, other comorbidities, and the physician’s choice. CGP-guided therapy with PARP inhibitors was administered to patients with BRCA 1 or BRCA 2 mutations after the initial response to standard-of-care first- or second-line systemic therapy, in accordance with the existing reimbursement restrictions for PARP inhibitors in Croatia as well as multidisciplinary team decisions.

### 2.3. Endpoints

The primary endpoint was to present and compare the proportion of patients carrying a BRCA 1 or BRCA 2 mutation with the proportion of patients having HRD or LOH, for which targeted therapy with PARP inhibitors was chosen.

Moreover, in order to further investigate the clinical value of comprehensive genomic profiling, we compared its results with the conventional testing for BRCA genes on 49 ovarian carcinoma patients in a single institution in the same period of two years, from January 2020 to December 2021. Conventional testing was performed either from blood or paraffin-embedded tissue.

Comprehensive genomic profiling is approved by FDA and has undergone many validations [23,24]. However, to confirm its results, we explored and compared its compatibility with locally performed BRCA testing and immunohistochemistry testing for TP53 mutations in patients at a single institution.

The secondary endpoint was to present other clinically relevant genomic alterations (CRGAs) detected, which were defined as those with approved targeted therapy for the patients’ tumor types or approved to treat other tumor types, or with existing clinical trials available. In addition, we conducted a comparison of the CGP results based on the histological subtypes.

### 2.4. Statistical Analysis

We describe the data as percentages, medians, and interquartile ranges (IQRs) using StataCorp 2019 software (Stata Statistical Software: Release 16, College Station, TX, USA: StataCorp LLC).

## 3. Results

### 3.1. Description of Patients and Previous Therapy

From 1 January 2020 to 1 December 2021, 86 patients with locally advanced or metastatic ovarian cancer were presented to multidisciplinary teams, and CGP was performed on their tumor tissue specimens. The median age was 59 (IQR 52–66) years, with a total range from 39 to 80 years (Table 1). The majority of patients, 71 (87%), were in good general condition with an ECOG performance status of 0. The most common histological subtype was high-grade serous cancer in 69 patients (80%). All patients received chemotherapy in either a neoadjuvant, adjuvant, or metastatic setting as a standard treatment for locally advanced metastatic ovarian cancer. Nineteen patients (22%) were newly diagnosed with metastatic ovarian cancer (Table 1). The median number of prior lines of therapy for metastatic disease was 1 (IQR 0–1) (Table 1). The most common chemotherapy protocol used as the first-line treatment was a combination of paclitaxel and carboplatin in 84% of patients.

### 3.2. Comprehensive Genomic Profiling

All 86 patients (100%) analyzed using CGP had at least one genomic alteration (GA). Clinically relevant genomic alterations (CRGAs) were detected in 73 patients (85%), with a median of 2 (IQR 1–3) CRGAs per patient (Table 2). The most common CRGA reported was the functional loss of the tumor suppressor p53 encoded by the TP53 gene, which was found in 48 patients (56%). The next most common CRGAs were those of phosphatide-inositol-3 kinases (PIK3), which were found in 14 patients (17%); CRGAs of KRAS (Kirsten rat sarcoma virus), which were found in 13 patients (15%); and CRGAs of NF 1 (encodes neurofibromin) or NF 2 (encodes merlin) mutations, which were found in 10 patients (12%) (Table 2). Genomic alterations without clinical significance were detected in 69 patients (80%) with a median of 2 (IQR 1–3) GAs per patient. The microsatellite status was determined to be highly unstable in only one patient (0.01%) and was not determined in three patients (0.03%). The median tumor mutational burden (TMB) was 3 (IQR 0–4) mutations per megabase (Muts/Mb), with a total range of 0 to 18. A high TMB (≥10 Muts/Mb) was reported in only two patients (0.02%). The median loss of heterozygosity (LOH) was 14.6 (IQR 6.8–21.7), with 35 patients (41%) having an LOH ≥16. The LOH status was not determined for five patients (0.06%). We found a BRCA-positive status in 22 patients (26%), with 15 patients (17%) carrying the BRCA1 mutation and 7 patients (8%) carrying the BRCA2 mutation. Altogether, 18 of 22 patients (81.7%) carrying a BRCA mutation had an LOH ≥ 16.

#### 3.2.1. Difference in CGP Results Regarding Histological Types

Considering that 80% of patients had high-grade serous ovarian cancer, a subanalysis of CGP regarding the histological subtype was performed. Patients were separated into two groups: (1) high-grade serous and (2) low-grade serous and other histological types. A markedly lower prevalence of clinically relevant mutations was found among the second group, with a difference also noted in BRCA status and LOH (Table 3).

#### 3.2.2. Comprehensive Genomic Profiling vs. Conventional Testing

Conventional testing for BRCA was performed from blood or paraffin-embedded tissue. Comparing the CGP results with conventional testings performed in the same period of two years, from January 2020 to December 2021, in a single institution, a clinically relevant difference was found, with a higher number of patients having BRCA mutations after CGP analysis. Moreover, CGP provided information regarding LOH, resulting in 27% more patients in total who would potentially benefit from PARP inhibitors (Table 4).

For the same group of patients coming from a single institution, we performed internal validation of the CGP results through the determination of BRCA status and immunohistochemistry confirmation of TP53 status. BRCA status was determined locally for nine patients, and the matching with CGP results was 100%. By contrast, immunohistochemistry for TP53 was performed locally in 20 patients, with 18 of them (90%) having the same results as CGP. One patient had a local IHC-confirmed TP53 mutation but was negative on CGP, and the second patient had a positive CGP finding with negative IHC local status for TP53.

After an analysis of all CGP reports and all GA reports, some type of targeted therapy was chosen for 56 patients (65%). Targeted therapy approved for the patients’ tumor type (on-label therapy) was reported in 41 patients (48%), while targeted therapy approved for other tumor types based on patients’ GA (off-label therapy) was reported in 55 patients (64%). All the on-label alteration-driven therapies were included in the DNA repair mechanism with PARP inhibitors, such as olaparib, niraparib, and rucaparib. The same group of patients had the most common off-label therapy as well, which was the PARP inhibitor talazoparib. Moreover, the next most common alteration-driven off-label therapies were those encompassing PI3K/mTOR (phosphoinositide-3 kinase/mammalian target of rapamycin) and Ras/Raf/MEK (mitogen-activated protein kinase) mutations, with mTOR and MEK inhibitors being the most frequently used targeted therapy. GCP-guided targeted therapy with PARP inhibitors was administered to 14 patients (16%) on the basis of the indication, clinical need, MDT decision, and reimbursement status of the therapy.

## 4. Discussion

### 4.1. Summary of Main Results

The results from the CGP analysis in our study showed that all patients with locally advanced or metastatic ovarian cancer harbored at least one genomic alteration. Additionally, the molecular profile of our group of patients is similar to that of previous findings for ovarian cancer, particularly to The Cancer Genome Atlas (TCGA) comprehensive profiling (12). Conventional testing in the ovarian cancer diagnostic workup using single-target assays that detect only BRCA mutation would have potentially indicated targeted therapy with PARP inhibitors in 22 patients (26%) in our group of tested patients. The results presented in Table 4 from a single institution show that conventional testing is less sensitive than CGP, particularly regarding somatic mutations. By contrast, CGP performed using next-generation sequencing revealed a need for targeted therapy with PARP inhibitors in 35 patients (41%), resulting in a clinically significant number of patients who would potentially benefit from already approved treatment options. Furthermore, the results of the CGP analysis provided information on other potential targetable mutations and, as a result, led to the discovery of more patients who would potentially benefit from targeted therapy.

### 4.2. Results in the Context of Published Literature

As mentioned previously, significant advances have occurred with the modification of the surgical approach, the administration of TC chemotherapy, and the introduction of immunotherapy and targeted therapy [2,3,4,5,6,7,8]. Despite the aforementioned improvements, ovarian cancer treatment outcomes are still rather unsatisfactory compared with those for some other cancer types due to the diagnosis at advanced stages and the inherent biological specificities [25]. In recent years, a focus on medicine, particularly in oncology, has been placed on individual patients, emphasizing the need for treatment personalization. Precision medicine, in a full sense, comprises a timely and organized individual approach, an extensive and treatment-oriented diagnostic workup, and personalized therapy [26]. Hence, by providing more detailed insights into the specific tumor genes and “genomic scarring” of each patient, CGP represents the next step toward precision oncology and enables the potential discovery of the next breakthrough regarding targeted alteration-driven therapy for individuals and a possible significant effect on the outcomes for women diagnosed with advanced ovarian cancer.

Although NGS is already recognized as fundamental for precision medicine, its limitations, such as the interpretation of the results, have been described, and we are still on the quest to define its optimal position and application in daily clinical practice [27,28]. Subsequently, several trials involving targeted therapy were conducted and produced controversial results; some trials, such as MOSCATO and SHIVA, reported negative outcomes with targeted therapy, while several other trials obtained positive results, with the effects of targeted therapy on the investigated outcomes approximately doubled [29,30,31,32,33,34]. Meanwhile, due to its specific genetic background, more than 15% of ovarian carcinoma patients carry germline mutations, and an additional 5–11% of patients carry somatic mutations in either the BRCA1 or BRCA2 gene. Including other HRD gene mutations, up to 50% of patients have HRD with existing excellent treatment opportunities with PARP inhibitors, making ovarian cancer an ideal tumor for an upfront CGP analysis [10,25]. In addition, targeted therapy, as well as obligatory biomarker detection as a part of the diagnostic process, has already been established in ovarian cancer management [25,35]. As a result, the European Society for Medical Oncology (ESMO) recommends the routine use of NGS in the ovarian cancer workup [36].

Furthermore, one can argue that with more precise diagnostics, classical clinical trials will no longer suffice for drawing conclusions regarding treatment, and we will have to learn for and from every patient individually. Hence, importance and emphasis are placed on real-world data and clinical experience to determine the real efficacy and toxicity, for which we already have positive feedback for the use of PARP inhibitors for ovarian cancer [37,38].

### 4.3. Strengths and Weaknesses

The limitations of our study, which may affect the interpretation of the results, include its retrospective design and the relatively small number of patients. The retrospective design of the study may lead to patient selection and subsequent bias in the study results.

On the other hand, this study presents the national experience of all consecutive patients tested, and our results define the potential importance of CGP in the daily clinical practice of treating patients with ovarian cancer.

### 4.4. Implications for Practice and Future Research

The study results present real-world data on a national level and, as such, have great validity for clinical practice and for setting the position of CGP analysis in everyday diagnostic and treatment management of locally advanced ovarian cancer. In addition, they bring a new perspective of personalized and precise treatment, in which therapy is tailored individually to patients according to their CGP findings. Hence, new research in this field could result in the development of novel treatment strategies.

## 5. Conclusions

Here, we presented the two-year experience of CGP in the ovarian cancer diagnostic workup. On the basis of the results, which indicate that a significantly higher number of women would achieve a possible benefit from targeted therapy, CGP should be integrated into the diagnostic workup of locally advanced and metastatic ovarian cancer as a backbone diagnostic tool.

## Figures and Tables

**Table 1 jpm-12-01176-t001:** Characteristics of patients before comprehensive genomic profiling.

	All Patients (n = 86)	
Age at the time of diagnosis, median (IQR)	52	(52–66)
Metastatic disease at the initial diagnosis	19	(22)
FIGO stage at diagnosis †		
I *	3	(3)
II *	6	(7)
III	58	(67)
IV	19	(22)
Histological subtypes †		
Serous carcinoma		
Low-grade	8	(9)
High-grade	69	(80)
Carcinosarcoma	2	(2)
Microcellular carcinoma	2	(2)
Clear cell carcinoma	1	(1)
Mixed types		
Endometrial + clear cell carcinoma	1	(1)
Granulosa cell tumor	1	(1)
Steroid cell tumor	1	(1)
Malignant seal ring cells	1	(1)
Number of patients receiving previous chemotherapy		
Neoadjuvant	18	(21)
Adjuvant	49	(57)
Number of previous treatment lines for Metastatic disease		
0	43	(50)
1	26	(30)
2	14	(16)
3	2	(2)
7	1	(1)
ECOG performance status before CGP		
0	71	(83)
1	13	(15)
Not determined	2	(2)

Data are presented as numbers (percentages) of patients unless stated otherwise. Abbreviations: IQR, interquartile range; CGP, comprehensive genomic profiling. Data were missing for the date of metastatic disease and number of previous treatment lines for metastatic disease in 1 patient (3%). † The total is <100% due to a rounding error. * CGP was performed upon progression.

**Table 2 jpm-12-01176-t002:** The results of comprehensive genomic profiling.

	All Patients (n = 86)	
Genomic alterations		
Any genomic alteration	86	(100)
Clinically relevant	73	(85)
Clinically not relevant	69	(80)
Number of genomic alterations, median (IQR)		
Total number	2	(1–3)
Clinically relevant	2	(1–3)
Not clinically relevant	2	(1–3)
Number of clinically relevant genomic alterations †		
0	13	(15)
1	20	(23)
2	20	(23)
3	14	(16)
4	8	(9)
5	11	(13)
Clinically relevant genomic alterations		
BRCA	22	(25)
BRCA 1	15	(17)
BRCA 2	7	(8)
TP53	48	(56)
PIK3 pathway	14	(17)
KRAS	13	(15)
NF 1/2	10	(12)
MYC	9	(10)
SOX2	7	(8)
PTEN or FGFR 1/2	5	(6)
CCND1/2 or AKT2 or ARID1A	4	(5)
CHEK2 or TSC1/2 or ERBB2	3	(4)
PDGFR A/B or AURKA or MDM2 or MET or ATM or NRAS or CDK12 or STK11	2	(2)
RICTOR or PALB2 or SMARCA4 or CTNNB1 or PTCH1 or BRAF or MTAP or AXL or MAP2K1 or KIT or NTRK2 or SMO	1	(1)
Loss of heterozygosity (LOH)		
Median (IQR)	14.6	(6.8–21.7)
LOH ≥ 16	35	(41)
Not determined	5	(6)
Microsatellite status		
Stable	82	(95)
High instability	1	(1)
Not determined	3	(4)
Tumour mutational burden (TMB), median (IQR)	3	(0–4)
Tumour mutational burden (TMB)		
Not high	81	(94)
High (≥10 mutations/Mb)	2	(2)
Not determined	3	(4)

Data are presented as numbers (percentages) of patients unless stated otherwise. Abbreviations: CGP, comprehensive genomic profiling; IQR, interquartile range. † The total is <100% due to a rounding error.

**Table 3 jpm-12-01176-t003:** Difference in CGP results regarding histological types.

	High-Grade Serous (n = 69)		Low-Grade Serous + Other Types
			(n = 17)
Genomic alterations			
Any genomic alteration	69	(100)	17(100)
Clinically relevant	61	(88)	12(71)
Not clinically relevant	65	(94)	12(71)
BRCA	21	(30)	1(6)
BRCA 1	14	(67)	1(6)
BRCA 2	7	(33)	0(0)
Loss of heterozygosity (LOH)			
Median (IQR)	16.4	(11.6–22.5)	2(0.5–6.8)
LOH ≥ 16	34	(49)	1(7)
Not determined	2	(3)	3(18)

Data are presented as numbers (percentages) of patients unless stated otherwise. Abbreviations: CGP, comprehensive genomic profiling; IQR, interquartile range.

**Table 4 jpm-12-01176-t004:** Comprehensive genomic profiling vs. conventional testing for BRCA.

	CGP Results (n = 33)		Conventional Testing
			(n = 49)
Testing from blood	0	(0)	31(63)
Testing from tissue	33	(100)	18(37)
BRCA	12	(36)	9(18)
BRCA 1	9	(27)	7(14)
BRCA 2	3	(9)	2(4)
Loss of heterozygosity (LOH)			
median (IQR)	15.7	(8.85–21.9)	
LOH ≥ 16	15	(45)	
not determined	2	(6)	49(100)

Data are presented as numbers (percentages) of patients unless stated otherwise. Abbreviations: CGP, comprehensive genomic profiling; IQR, interquartile range.

## Data Availability

The data used to support the findings of this project are available from the corresponding author upon request.

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
