# Peer review of "Comprehensive Genomic Profiling in the Management of Ovarian Cancer—National Results from Croatia"

_jpm, 2022, doi:10.3390/jpm12071176_

Round 1
Reviewer 1 Report
Precision medicine is up-to-date evidenced based medicine. But not all patients not all country provide equal accessibility in terms of diagnosis and treatment.
1. What are the unique characteristics of Croatia and its ovary cancer patients? Incidence, mortality target treatment reimbursement compared to other European country.
2. Comprehensive genomic profiling experience is incredibly special but there should be characteristic figure about this population compared to other ovary cancer patient in Croatia. In addition, Foundation CDx is already adopted in several study population and real-world patients in some countries. What’s specific result in this study population?
3. ASCO/SGO/NCCN guidelines are recommendable but insurance/coverage problem in country/institution are different problem. What’s availability of Sanger sequencing and PARP inhibitor/Avastin drug coverage in Croatia? Was there founder mutation in Croatian people?
Author Response
Precision medicine is up-to-date evidenced based medicine. But not all patients not all country provide equal accessibility in terms of diagnosis and treatment.
[1] What are the unique characteristics of Croatia and its ovary cancer patients? Incidence, mortality target treatment reimbursement compared to other European country.
Response: Thank you for your dedicated time and revision. Thank you for this constructive comment, we have compared the incidence in Croatia and other European countries and described the treatment strategies in Croatia, and have incorporated it as a part of Introduction, Line 83-94.
Now, the text of that part of introduction states:
„Ovarian cancer represents one of the major health burdens in Croatia due to its high mortality to incidence ratio (0,67), which puts Croatia among countries with highest mortality and incidence in Europe [18, 19]. Potential reason for high mortality to incidence ratio lays in the late diagnosis and lack of proper treatments. For instance, in 2018, Croatia was one of the countries with lowest tier for PARP inhibitors uptake [19]. Ovarian cancer patients are treated with standard chemotherapy following surgery (or before when neoadjuvant therapy is indicated) or in the case of initially metastatic disease, with platinum-based chemotherapy and paclitaxel three-weekly or dose dense, +/- bevacizumab or from recently PARP inhibitor, depending on the residual disease and BRCA status as well as response to platinum therapy. In the treatment of recurrent disease, patients are also treated with standard chemotherapy based on platinum sensitivity, along with bevacizumab or with PARP inhibitors in case of a BRCA mutation. PARP inhibitors are given after response to platinum based chemotherapy as a maintenance treatment. However, they are not indicated in case of HRD or LOH like in some other European Countries.“
[2] Comprehensive genomic profiling experience is incredibly special but there should be characteristic figure about this population compared to other ovary cancer patient in Croatia. In addition, Foundation CDx is already adopted in several study population and real-world patients in some countries. What’s specific result in this study population?
Response: Thank you once again for constructive inputs. This is, in our knowledge, the first study of CGP of ovarian cancer on a whole country population (of course of those patients who fulfilled defined criteria from Methods, Line 144-146). Routine conventional testing with NGS is performed only for high-grade serous ovarian cancers in two centers in Croatia. Until recently, it was only performed for platinum-sensitive patients, but since the reimbursement of PARP inhibitors in first line, it is performed in all patients regardless of platinum-sensitivity.
Specificity of our results is significant number of patients who would benefit from PARP inhibitors because of analysis of loss of heterozygosity from CGP. Also, in order to compare and validate conventional BRCA testing we have performed analysis in a single institution and found higher prevalence of BRCA mutation with CGP. Hence, we have included it in the Results, Line 234-239, Table 4.
Now, the text of that part of results states:
„Conventional testing for BRCA is performed from blood or paraffin-embedded tissue analysis. Comparing the CGP results with conventional testings performed in the same period of two years, from January 2020 to December 2021, in a single institution, the clinically relevant difference was found with higher number of patients having BRCA mutations after CGP analysis. Moreover, CGP provided information regarding LOH, resulting in 27% more patients in total who would potentially have benefit from PARP inhibitors, Table 4.“
Table 4. Comprehensive genomic profiling vs. conventional testing for BRCA
|
|
CGP results (n=33) |
Conventional testing |
|
|
|
(n=49) |
|
|||
|
Testing from blood |
0 |
(0) |
31 (63) |
|
|
Testing from tissue |
33 |
(100) |
18 (37) |
|
|
BRCA |
12 |
(36) |
9 (18) |
|
|
BRCA 1 |
9 |
(27) |
7 (14) |
|
|
BRCA 2 |
3 |
(9) |
2 (4) |
|
|
|
|
|
|
|
|
Loss of heterozygosity (LOH) |
|
|
|
|
|
median (IQR) |
15.7 |
(8.85-21.9) |
|
|
|
LOH ≥ 16 |
15 |
(45) |
|
|
|
not determined |
2 |
(6) |
49 (100) |
|
|
|
|
|
|
|
Data are presented as the numbers (percentages) of patients if not stated otherwise.
Abbreviations: CGP, comprehensive genomic profiling; IQR, interquartile range
[3] ASCO/SGO/NCCN guidelines are recommendable but insurance/coverage problem in country/institution are different problem. What’s availability of Sanger sequencing and PARP inhibitor/Avastin drug coverage in Croatia? Was there founder mutation in Croatian people?
Response: Thank you for the helpful and insightful remarks. While NGS (Ilumina) is available and while the price of the same is covered by health insurance (and sponsored by pharma company that are distributing PARP inhibitors) they are covering just germline mutation status and based on that we are missing patients with somatic mutations and with LOH for specific, biomarker selected treatment with PARP inhibitors. Fortunately, Croatia is one of the few countries that has CGP available on a national level and special fund for treatment and with that we can significantly increase number of patients, particularly ovarian cancer patients, who can be treated with targeted therapy.
Reimbursements of therapies are described in part of introduction, line 86-94.
We are not aware of any specific founder mutation in Croatia population.
Furthermore, we did not mention it in our manuscript but in 2017 In the Laboratory for Hereditary Cancer at the Rudjer Boskovic Institute the first case report of BRCA1 exons 4-6 deletion was determined and sequencing confirmed the deletion as NG_005905.2:g.107648_117905del10257 (ref Musani V, Sušac I, Ozretić P, Eljuga D, Levanat S. The first case report of a large deletion of the BRCA1 gene in Croatia: A case report. Medicine (Baltimore). 2017 Dec;96(48):e8667. doi: 10.1097/MD.0000000000008667. PMID: 29310340; PMCID: PMC5728741.).

Reviewer 2 Report
In this manuscript by Cerina et al, the authors have reported CGP analysis of 86 ovarian cancer patients and highlighted clinically relevant genomic alterations. However, few questions need to be addressed.
1. Specifically which mutations were found in the altered genes? 2. Out of 86 patients, 69 were high-grade serous cancer patients. Was there any significance difference of CGP analysis/ gene alterations in high-grade patients compared to rest of the patients? 3. All these genetic alterations need to be validated at least in a few patients to confirm the CGP analysis. IHC should be done in patient samples to show any differential expression patterns of genetic alterations among ovarian cancer patients. 4. The methodology of CGP should be mentioned in the method section, detail procedure.Author Response
In this manuscript by Cerina et al, the authors have reported CGP analysis of 86 ovarian cancer patients and highlighted clinically relevant genomic alterations. However, few questions need to be addressed.
[1] Specifically which mutations were found in the altered genes?
Response: Thank you for dedicated time and revision. All altered genes are listed in Table 2. However, mutations of each altered gene for all patients would be hard to sum up or put in the table but we have relied on their clinical significance considering that FMI collaborates with Flatiron health and Biopharma (as well as with other repositories) and that all reports are updated accordingly to the latest findings and FDA approvals. In addition, for purposes of this research, most common BRCA 1 mutation were Q1756fs*74 in 3 patients and S282fs*15 in 2 patients and BRCA 2 N31241* in 2 patients.
[2] Out of 86 patients, 69 were high-grade serous cancer patients. Was there any significance difference of CGP analysis/ gene alterations in high-grade patients compared to rest of the patients?
Response: Thank you for very interesting point. We have compared the groups and presented the results of analysis in the “Results”; Line 222-226 and Table 3.
Now, the text of that part of results states:
„Considering that 80% of patients had high-grade serous ovarian cancer, subanalysis of CGP regarding the histological subtype was performed. Patients were separated into two groups, high-grade serous vs low-grade serous and other histological types. Markedly lower prevalence of clinically relevant mutations was found among the second group with also noted difference in BRCA status and LOH, Table 3.“
Table 3. Difference of CGP results regarding histological types
|
|
High-grade serous (n=69) |
Low-grade serous + other types |
|
|
|
(n=17) |
|
|||
|
Genomic alterations |
|
|
|
|
|
any genomic alteration |
69 |
(100) |
17 (100) |
|
|
clinically relevant |
61 |
(88) |
12 (71) |
|
|
clinically not relevant |
65 |
(94) |
12 (71) |
|
|
|
|
|
|
|
|
BRCA |
21 |
(30) |
1 (6) |
|
|
BRCA 1 |
14 |
(67) |
1 (6) |
|
|
BRCA 2 |
7 |
(33) |
0 (0) |
|
|
|
|
|
|
|
|
Loss of heterozygosity (LOH) |
|
|
|
|
|
median (IQR) |
16.4 |
(11.6-22.5) |
2 (0.5-6.8) |
|
|
LOH ≥ 16 |
34 |
(49) |
1 (7) |
|
|
not determined |
2 |
(3) |
3 (18) |
|
|
|
|
|
|
|
Data are presented as the numbers (percentages) of patients if not stated otherwise.
Abbreviations: CGP, comprehensive genomic profiling; IQR, interquartile range
[3] All these genetic alterations need to be validated at least in a few patients to confirm the CGP analysis. IHC should be done in patient samples to show any differential expression patterns of genetic alterations among ovarian cancer patients.
Response: Thank you for the valuable observation. Considering our analysis is retrospective in nature and on a national level with ethical committee approval for CGP analysis only, we did not validate its results. However, certain percent of patients did have locally confirmed BRCA status (either from blood or paraffin-embedded tissue) and the results were matching with CGP reports. Considering that vast majority of patients did not have that, we did not mention it in the manuscript. Regarding IHC as a part of standard diagnostic routine for ovarian cancer, we can only perform P53 analysis for high grade serous ovarian cancer and we didn´t check but most likely it was done for majority of patients. Also, IHC could be potentially performed for MYC, ERBB2, c-kit and NTRK but it is not part of routine RT-PCR diagnostics. As for the low grade serous ovarian cancer we have several KRAS mutations confirmed on IHC and could probably perform IHC for PIK3 and BRAF but it is also not standard practice. We do not have IHC options for LOH and TMB.
[4] The methodology of CGP should be mentioned in the method section, detail procedure.
Response: Thank you once more for the constructive and helpful comment. We absolutely agree with you so we have incorporated it in the “Methods” under “Project design”, Lines 112-131.
Now, the text of that part of methods states:
„Formalin-fixed, paraffin-embedded tissue was sent as a block and one hematoxylin and eosin stained slide or 10 unstained slides with one hematoxylin and eosin stained slide. Minimal surface area was 25 mm2 and minimal tumor content was 20%, while optimal was 30% of tumor nuclei, defined as the number of tumor cells divided by total number of all cells with nuclei. Once the DNA was extracted, 50-1000 ng underwent whole-genome shotgun library construction and hybridization-based capture in order to detect alterations of 324 genes in total, of which 309 exons related with tumors, one promoter region, one non-coding RNA and certain regions of introns in 34 frequently rearranged genes in tumors. Illumina® HiSeq 4000 was used to sequence hybrid capture-selected libraries to high uniform depth. The typical median depth of coverage was >500x with >99% of exons at coverage >100x. The sequenced regions were analyzed for four different types of alterations; base substitution, deletion or insertion, copy number variation and gene redistribution in a group of genes associated with the tumor development. The microsatellite status was based on genome wide analysis of 95 microsatellite loci, while TMB was determined by counting all synonymous and non-synonymous variants present at 5% allele frequency or greater and total number was presented as mutations per megabase (Muts/Mb) unit, while homologous recombination repair (HRR) mechanism is assessed for mutations in the 14 HRR genes, ATM, BARD1, BRCA1, BRCA2, BRIP1, CDK12, CHEK1, CHEK2, FANCL, PALB2, RAD51B, RAD51C, RAD51D, and RAD54L [17].“

Round 2
Reviewer 1 Report
Well corrected and added
Thanks
Author Response
Thank you for acceptance of all of our answers and acceptance of the revised version of article for publication.

Reviewer 2 Report
The authors have partly answered to a few questions. However, it is important and necessary to validate genetic alterations in high-grade serous cancer patients considering the paraffin tissue sections are available, otherwise results might be misleading.
Author Response
The authors have partly answered to a few questions. However, it is important and necessary to validate genetic alterations in high-grade serous cancer patients considering the paraffin tissue sections are available, otherwise results might be misleading.
Response: Thank you for your dedicated time and fast revision. Also, thank you for your all constructive suggestions so far. As requested, we have performed local validation of CGP results (BRCA by NGS and TP53 by IHC) and consequently have introduced new sentence in the methods; lines 172-180; 184, and results as a part of comparison analysis; lines 267-271.
Now that part of text states:
Methods:
“Moreover, in order to further investigate clinical value of comprehensive genomic profiling, we have compared its results with the conventional testing for BRCA genes on 49 ovarian carcinoma patients in a single institution in the same period of two years, from January 2020 to December 2021. Conventional testing was performed either from blood or paraffin-embedded tissue.
Comprehensive genomic profiling is approved by FDA and has underwent many validations [23, 24]. However, to confirm its results, we have explored and compared their compatibility with locally performed BRCA testing and immunohistochemistry testing for TP53 mutation, among patients from a single institution.”
“Also, we did comparison of the CGP results based on the histological subtypes.“
Results:
“For the same group of patients coming from a single institution, we have performed internal validation of CGP results through determination of BRCA status and immunohistochemistry confirmation of TP53 status. BRCA status was determined locally for 9 patients and matching with CGP results was 100%. While, immunohistochemistry for TP53 was performed locally in 20 patients with 18 of them (90%) having same results as CGP. One patients had locally IHC confirmed TP53 mutation but negative on CGP and second patient had positive CGP finding with negative IHC local status for TP53.“
